# DENSE MORPHOLOGICAL NETWORK: AN UNIVERSAL FUNCTION APPROXIMATOR

## ABSTRACT

Artificial neural networks are built on the basic operation of linear combination and non-linear activation function. Theoretically this structure can approximate any continuous function with three layer architecture. But in practice learning the parameters of such network can be hard. Also the choice of activation function can greatly impact the performance of the network. In this paper we are proposing to replace the basic linear combination operation with non-linear operations that do away with the need of additional non-linear activation function. To this end we are proposing the use of elementary morphological operations (dilation and erosion) as the basic operation in neurons. We show that these networks (Denoted as DenMo-Net) with morphological operations can approximate any smooth function requiring less number of parameters than what is necessary for normal neural networks. The results show that our network perform favorably when compared with similar structured network.

## 1 INTRODUCTION

In artificial neural networks, the basic building block is an artificial neuron or perceptron that simply computes the linear combination of the input (Rosenblatt, 1958). It is usually followed by a non-linear activation function to model the non-linearity of the output. Although the neurons are simple in nature, when connected together they can approximate any continuous function of the input (Hornik, 1991). This has been successfully utilized in solving different real world problems like image classification (Krizhevsky et al., 2012), semantic segmentation (Long et al., 2015) and image generation (Isola et al., 2017). While these models are quite powerful in nature, their efficient training can be hard in general (LeCun et al., 2012) and they need support of specials techniques, such as batch normalization (Ioffe & Szegedy, 2015) and dropout (Srivastava et al., 2014), in order to achieve better generalization capabilities. Their training time also depends on the choice of activation function (Mishkin et al., 2017).

In this paper we are proposing new building blocks for building networks similar to neural network. Here, instead of the linear combination operation of the artificial neurons, we use a non-linear operation that eliminates the need of additional activation function while requiring a small number of neurons to attain same performance or better. More specifically, We use morphological operations (i.e. dilation and erosion) as the elementary operation of the neurons in the network. Our contribution in this paper is building a network with these operations that has the following properties.

1. Networks built with with dilation-erosion neurons followed by linear combination can approximate any continuous function given enough dilation/erosion neurons.
2. As dilation and erosion operation are non-linear by themselves, requirement of separate non-linear activation function is eliminated.
3. The use of dilation-erosion operation greatly increases number of possible decision boundaries. As a result, complex decision boundaries can be learned using small number of parameters.

The rest of the paper is organized as follows. Section 2 describes the prior work on morphological neural network. In Section 3, we introduce our proposed network and prove its capabilities theoretically. We further demonstrate its capabilities empirically on a few benchmark datasets in Section 4. Lastly Section 6 concludes the paper.

## 2 RELATED WORK

Morphological neuron was first introduced by Davidson & Hummer (1993) in their effort to learn the structuring element of dilation operation in images. Similar effort has been made to learn the structuring elements in a more recent work by Masci et al. (2013). Use of morphological neurons in a more general setting was first proposed by Ritter & Sussner (1996). They restricted the network to a single layer architecture and focused only on binary classification task. To classify the data, these networks use two axis parallel hyperplanes as the decision boundary. This single layer architecture of Ritter & Sussner (1996) has been extended to two layer architecture by Sussner (1998). This two layer architecture is able to learn multiple axis parallel hyperplanes, and therefore is able to solve arbitrary binary classification task. But, in general the decision boundaries may not be axis parallel, as a result this two layer network may need to learn a large number of hyperplanes to achieve good results. So, one natural extension is to incorporate the option to rotate the hyperplanes. Taking a cue from this idea, Barmpoutis & Ritter (2006) proposed to learn a rotational matrix that rotates the input before trying to classify the data using axis parallel hyperplanes. In a separate work by Ritter et al. (2014) the use of $L^1$ and $L^\infty$ norm has been proposed as a replacement of the *max/min* operation of dilation and erosion in order to smooth the decision boundaries.

Ritter & Urcid (2003) first introduced the dendritic structure of biological neurons to the morphological neurons. This new structure creates hyperbox based decision boundaries instead of hyperplanes. The authors have proved that with hyperboxes any compact region can be estimated, therefore any two class classification problems can be solved. A generalization of this structure to the multiclass case has also been done by (Ritter & Urcid, 2007). Sussner & Esmi (2011) had proposed a new type of structure called morphological perceptrons with competitive neurons, where the output is computed in winner-take-all strategy. This is modelled using the *argmax* operator and this allows the network to learn more complex decision boundaries. Later Sossa & Guevara (2014) proposed a new training strategy to train this model with competitive neurons.

The non-differentiability of the max-min operations has forced the researchers to propose specialized training procedures for their models. So, a separate line of research has attempted to modify these networks so that gradient descent based optimizer can be used for training. Pessoa & Maragos (2000) have combined the classical perceptron with the morphological perceptron. The output of each node is taken as the convex combination of the classical and the morphological perceptron. Although max/min operation is not differentiable, they have proposed methodology to circumvent this problem. They have shown that this network can perform complex classification tasks. Morphological neurons have also been employed for regression task. de A. Arajo (2012) has utilized network architecture similar to morphological perceptrons with competitive learning to forecast stock markets. The *argmax* operator is replaced with a linear function so that the network is able to regress forecasts. The use of linear activation function enables the use of gradient descent for training which is not possible with the *argmax* operator. For morphological neurons with dendritic structure Zamora & Sossa (2017) had proposed to replace the *argmax* operator with a softmax function. This overcomes the problem of gradient computation and therefore gradient descent is employed to train the network. So, this retains the hyperbox based boundaries of the dendritic networks, but facilitates easy training with gradient descent.

## 3 DENSE MORPHOLOGICAL NETWORK

In this section we introduce the basic components and structure of our network and establish its approximation power.

### 3.1 DILATION AND EROSION NEURONS

Dilation and Erosion are two basic operations of our proposed network. Given an input $\boldsymbol{x} \in \mathbb{R}^d$ and some structuring element $\boldsymbol{s} \in \mathbb{R}^{d+1}$, **dilation** ($\oplus$) and **erosion** ($\ominus$) neurons computes the following two functions respectively

$$\boldsymbol{x} \oplus \boldsymbol{s} = \max_k(x'_k + s_k), \tag{1}$$

$$\boldsymbol{x} \ominus \boldsymbol{s} = \min_k(x'_k - s_k). \tag{2}$$

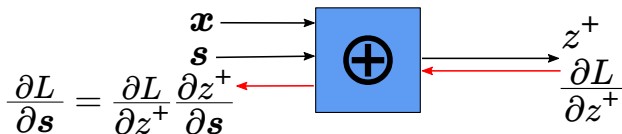

$$\frac{\partial L}{\partial \boldsymbol{s}} = \frac{\partial L}{\partial z^+}\frac{\partial z^+}{\partial \boldsymbol{s}}$$

Figure 1: Forward and backward pass through a node computing dilation

Where $\boldsymbol{x}' = [\boldsymbol{x}, 0]$ and $x'_k$ denotes the $k^{th}$ component of vector $\boldsymbol{x}'$. The 0 is appended to the input $\boldsymbol{x}$ to take care of the 'bias'. Here we try to learn the structuring element ($\boldsymbol{s}$). Note that erosion operation can also be written in the following form.

$$\boldsymbol{x} \ominus \boldsymbol{s} = -\max_k(s_k - x'_k) \tag{3}$$

### 3.2 Gradient of Dilation and Erosion

Artificial neural networks are trained using back-propagation. To be able to use the dilation/erosion neurons as a drop-in replacement of the artificial neurons, we must be able to compute the gradient of this operation. Here we show the gradient of the dilation operation. For erosion the gradient computation will be similar. Figure 1 shows the computational graph model of dilation operation and its gradient flow. Lets assume $L$ is the loss, we are trying to optimize. We need to compute $\frac{\partial L}{\partial \boldsymbol{s}}$ to be able to update the structuring element. This can be computed using the chain rule as follows.

$$\frac{\partial L}{\partial \boldsymbol{s}} = \frac{\partial L}{\partial z^+}\frac{\partial z^+}{\partial \boldsymbol{s}}. \tag{4}$$

Here $\frac{\partial L}{\partial z^+}$ will be the input gradient to this node which will be available from the node following this node. We need to compute $\frac{\partial z^+}{\partial \boldsymbol{s}}$. Now as $\boldsymbol{s} \in \mathbb{R}^{d+1}$ we can write the following.

$$\frac{\partial z^+}{\partial \boldsymbol{s}} = [\frac{\partial z^+}{\partial s_1}, \frac{\partial z^+}{\partial s_2}, \cdots, \frac{\partial z^+}{\partial s_{d+1}}]^T. \tag{5}$$

Now for each $\frac{\partial z^+}{\partial s_i}$ the gradient is computed as follows,

$$\frac{\partial z^+}{\partial s_i} = \begin{cases} 1 & \text{if } \boldsymbol{x} \oplus \boldsymbol{s} = x'_i + s_i, \\ 0 & \text{otherwise.} \end{cases} \tag{6}$$

### 3.3 Network Structure

The Dense Morphological Net or 'DenMo-Net', in short, that we propose here is a simple feed forward network with some dilation and erosion neurons followed by classical artificial neurons (Figure 2). We call the layer of dilation and erosion neurons as the **dilation-erosion layer** and the following layer as the **linear combination layer**. Let's assume the dilation-erosion layer contains $n$ dilation neurons and $m$ erosion neurons, followed by $c$ neurons in the linear combination layer. Let $\boldsymbol{x} \in \mathbb{R}^d$ is the input to the network. Let $z_i^+$ and $z_j^-$ be the output of $i^{th}$ dilation neuron and $j^{th}$ erosion node, respectively. Then we can write,

$$z_i^+ = \boldsymbol{x} \oplus \boldsymbol{s}_i^+, \tag{7}$$
$$z_j^- = \boldsymbol{x} \ominus \boldsymbol{s}_j^- \tag{8}$$

where, $\boldsymbol{s}_i^+$ and $\boldsymbol{s}_j^-$ are the structuring elements of the $i^{th}$ dilation neuron and $j^{th}$ erosion neuron respectively. Note that $i \in \{1, 2, \ldots, n\}$ and $j \in \{1, 2, \ldots, m\}$. The final output from a node of the linear combination layer is computed in the following way.

$$g(\boldsymbol{x}) = \sum_{i=1}^{n} z_i^+ \omega_i^+ + \sum_{j=1}^{m} z_j^- \omega_j^- \tag{9}$$

where $\omega_i^+$ and $\omega_j^-$ are the weights of the artificial neuron in the linear combination layer. In following subsection we show that $g(\boldsymbol{x})$ can approximate any continuous function $f : \mathbb{R}^d \to \mathbb{R}$.

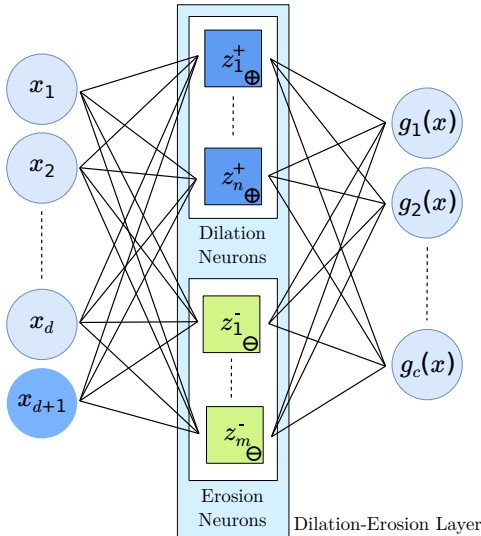

Figure 2: Single Layer DenMo-Net with $n$ dilation and $m$ erosion neuron and $c$ output neurons

### 3.4 FUNCTION APPROXIMATION

Here we show that with the linear combination of dilation and erosion, any function can be approximated, and the approximation error decreases with increase in the number of neurons in the dilation-erosion layer. Before that we need to describe some concepts.

**Definition 1 ($k$-order Hinge Function (Wang & Sun, 2005))** *A $k$-order hinge function consists of $(k+1)$ hyperplanes continuously joined together. it is defined by the following equation,*

$$h^{(k)}(\boldsymbol{x}) = \pm \max\{\boldsymbol{w}_1^T\boldsymbol{x} + b_1, \boldsymbol{w}_2^T\boldsymbol{x} + b_2, \ldots, \boldsymbol{w}_{k+1}^T\boldsymbol{x} + b_{k+1}\}. \tag{10}$$

**Definition 2 ($d$-order hinging hyperplanes ($d$-HH) (Wang & Sun, 2005))** *A $d$-order hinging hyperplanes ($d$-HH) is defined as the sum of multi-order hinge function as follows,*

$$\sum_i \alpha_i h^{(k_i)}(\boldsymbol{x}) \tag{11}$$

*with $\alpha_i \in \{-1, 1\}$, $k_i \leq d$.*

From Wang & Sun (2005) the following can be said about hinging hyperplanes.

**Proposition 1** *For any given positive integer $d$ and arbitrary continuous piece-wise linear function $f : \mathbb{R}^d \to \mathbb{R}$, there exists finite, say $N$, positive integers $\eta(k) \leq d+1, 1 \leq k < N$ and corresponding $\alpha_i \in \{-1, 1\}$ such that*

$$f(\boldsymbol{x}) = \sum_{k=1}^{N} \alpha_i h^{(\eta(k))}(\boldsymbol{x}), \quad \forall \boldsymbol{x} \in \mathbb{R}^d. \tag{12}$$

This says that any continuous piece-wise linear function of $d$ variables can be written as an $d$-HH, i.e. the sum of multi-order hinge functions. Now to show that our network can approximate any continuous functions, we show the following.

**Lemma 1** *$g(\boldsymbol{x})$ is sum of multi-order hinge functions.*

The proof of this lemma is given in Appendix A. Basically we show that $g(\boldsymbol{x})$ can written as the sum of $l$ hinge functions in the following form.

$$g(\boldsymbol{x}) = \sum_{i=1}^{l} \alpha_i \phi_i(\boldsymbol{x}) \tag{13}$$

where $l = m + n$ (number of neurons in the dilation-erosion layer), $\alpha_i \in \{1, -1\}$ and $\phi_i(\boldsymbol{x})$'s are $d$-order hinge function.

**Proposition 2 (Stone-Weierstrass approximation theorem)** *Let $C$ be a compact domain ($C \subset \mathbb{R}^d$) and $f : C \to \mathbb{R}$ a continuous function. Then there exists a continuous piece wise linear function $g$ such that for all $\boldsymbol{x} \in C$, $|f(\boldsymbol{x}) - g(\boldsymbol{x})| < \epsilon$ for some $\epsilon > 0$.*

**Theorem 1 (Universal approximation)** *Only a single dilation-erosion layer followed by a linear combination layer can approximate any continuous smooth function provided there are enough nodes in dilation erosion-layer.*

**Sketch of Proof** From lemma 1 we know that our DenMo-Net with of $n$ dilation and $m$ erosion neurons followed by a linear combination layer computes $g(x)$, which is a sum of multi-order hinge functions. Now from proposition 1 we get that any continuous piecewise linear function can be written by a finite sum of multi-order hinge function. Now from Proposition 2 we can say that any continuous function can be well approximated by a piecewise linear function. In general if $l \to \infty$ then $\epsilon \to 0$. If we increase the number of neurons in the dilation-erosion layer the approximation error decreases. Therefore, we can say that a DenMo-Net with enough dilation and erosion neurons can approximate any continuous function.

### 3.5 Learned Decision Boundary

The DenMo-Net we have defined above learns the following function,

$$g(\boldsymbol{x}) = \sum_{i=1}^{l} \alpha_i \phi_i(\boldsymbol{x}). \tag{14}$$

Where each $\phi(\boldsymbol{x})$ is collection of multiple hyperplanes joined together. Therefore the number of hyperplanes learned by the network with $l$ neurons in the dilation-erosion layer is much more than $l$. Each morphological neuron allows only one of the inputs to pass through because of $\max / \min$ operation after addition with the structuring element. So, effectively each neuron in the dilation-erosion layer chooses one component of the $d$-dimensional input vector. Depending on which component is being chosen, the final linear combination layer computes the hyperplane by taking either all the components of the input or only some of them (when a subset of input components is chosen more than once in the dilation-erosion layer). Note that this choice depends on the input and the structuring element together. For a network with $d$ dimensional input data and $l$ neurons ($l \geq d$) in the dilation-erosion layer, theoretically $(d + 1)^l - 1$ hyperplanes can be formed in $d$ dimension. Out of the all possible planes only ${}^l P_d \times (d + 1)^{l-d}$ planes can span anywhere in the $d$ dimensional space. Therefore, increasing the number of neurons in the dilation-erosion layer exponentially increases the possible number of hyperplanes, i.e., the decision boundaries. This implies that, using only a small number of neurons, complex decision boundaries can be learned.

## 4 Results

Here we empirically validate the power of our DenMo-Net and demonstrate its advantages in comparison with other networks like artificial neural networks with different activation functions i.e. tanh (**NN-tanh**) and ReLU (**NN-ReLU**) and Maxout network (Goodfellow et al., 2013). As our network is defined with all possible connections between two consecutive layers, we have compared with only similar structured networks. We have chosen the maxout network for comparison, because it uses the *max* function as a replacement of the activation function but with added nodes to compute the maximum. The experiments have been carried out on a toy dataset with two concentric circles for visualizing the decision boundaries and also on benchmark datasets like MNIST (LeCun et al., 1998), Fashion-MNIST (Xiao et al., 2017), CIFAR-10 and CIFAR-100 (Krizhevsky & Hinton, 2009). For all the tasks we have used categorical cross entropy as the loss and in the last layer softmax function is used. In the training phase, all the networks have been optimized using Adam (learning rate= $0.001, \beta_1 = 0.9, \beta_2 = 0.999$) optimizer (Kingma & Ba, 2014) with mini batches of size 32. In all the experiments, we have used same number of dilation and erosion neurons in dilation-erosion layer unless otherwise stated.

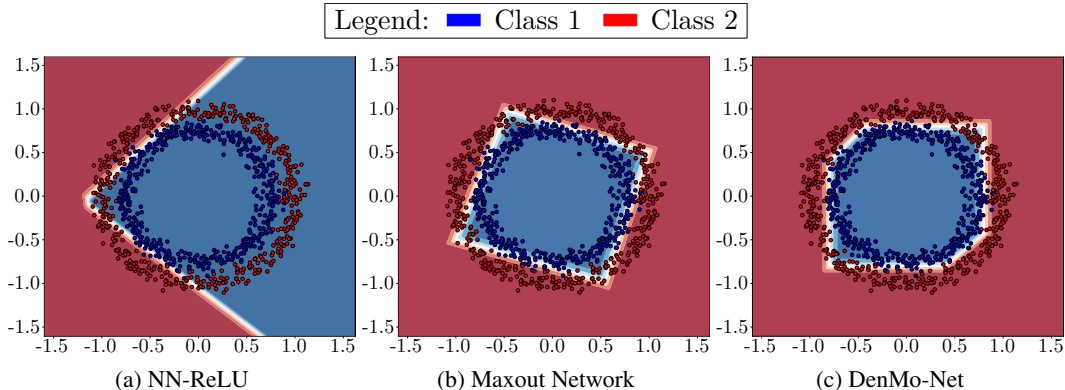

Figure 3: Decision boundaries of different networks

Table 1: Training accuracy achieved on the circle dataset by different networks

| Methods | Hidden nodes | Parameters | Training accuracy |
|---|---|---|---|
| NN-ReLU | 2 | 12 | 68.87 |
| NN-tanh | 2 | 12 | 69.10 |
| Maxout Network (h=2) | 2 | 18 | 87.17 |
| DenMo-Net | 2 | 12 | **91.6** |

## 4.1 VISUALIZATION WITH A TOY DATASET

For visualizing the decision boundaries learned by the classifiers, we have generated data on two concentric circles belonging to two different classes with center at the origin. We compare the results when only two neurons are taken in the hidden layer in all the networks. It is observed that classical neural network fails to classify this data with two hidden neurons as it learns one hyperplane per one hidden neuron. The boundaries learned by the network with ReLU activation function (NN-ReLU) is shown in figure 3a. The result of maxout network is better (87.17% training accuracy) as it introduces extra parameters with $\max$ function to achieve non-linearity. In the maxout layer we have taken maximum among $h = 2$ features. As we see in the figure 3b the network learns $(2 * h =)$ 4 straight lines when trying to classify these data. For the same data and two neurons in dilation-erosion layer, our DenMo-Net has learned 6 lines to form the decision boundary (figure 3c). Although from equation 14 we can say that we get at most 8 lines, only two of them can be placed anywhere in the 2D space while others are parallel to the axes. For this reason, we are getting two slanted lines and the remaining lines are parallel to the axes. The classification accuracy achieved by the networks along with their number of parameters is reported in table 1. The difference in the accuracy clearly shows the power of DenMo-Net.

## 4.2 MNIST DATASET

MNIST dataset (LeCun et al., 1998) contains gray scale images of hand written numbers (0-9) of size $28 \times 28$. It has 60,000 training images and 10,000 test images. Since our network does not support two dimensional input, we have converted each image to a column vector (in row major order) before giving it as input. The network we use follows the structure we have previously defined: input layer, dilation-erosion layer and linear combination layer computing the output. As in this dataset we had to distinguish between 10 classes of images, 10 neurons are taken in the output layer. In table 2 we have shown the accuracy on test data after training the network for 150 epochs with different number of nodes ($l$) in the dilation-erosion layer. The change of test accuracy over the epochs is shown in figure 4a. It is seen that increasing number of nodes in the dilation-erosion layer helps to increase non-linearity, and thus it results in better accuracy on test data. We get test average accuracy of 98.43% after training 3 times with the DenMo-Net of 200 dilation and 200 erosion neurons (Table 3) up to 400 epochs. We have also experimented when only dilation, only erosion and both type of neurons were used in the dilation-erosion layer (Figure 4b). We see that using both erosion only and both dilation and erosion neurons giving better accuracy.

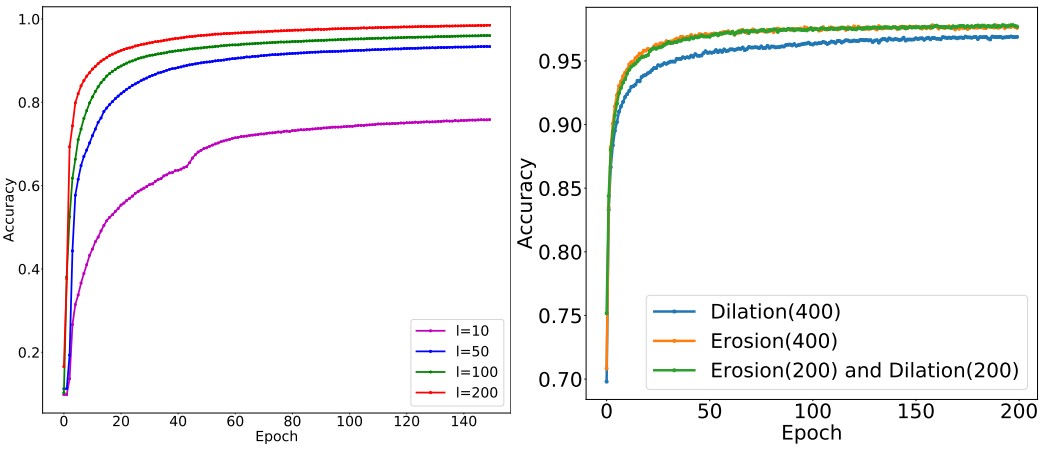

(a) Varying $l$ in the dilation-erosion layer  (b) Using only dilation, only erosion and both

Figure 4: Test accuracy achieved over epochs in the MNIST dataset

Table 2: Accuracy on MNIST dataset with different architectures

| Neurons in dilation-erosion layer ($l$) | 10 | 50 | 100 | 200 |
|---|---|---|---|---|
| Test Accuracy | 76.35 | 93.38 | 95.51 | 96.85 |

### 4.3 FASHION-MNIST DATASET

The Fashion-MNIST dataset (Xiao et al., 2017) has been proposed with the aim of replacing the popular MNIST dataset. Similar to the MNIST dataset this also contains $28 \times 28$ images of 10 classes and 60,000 training and 10,000 testing samples. While MNIST is still a popular choice for benchmarking classifiers, the authors' claim that MNIST is too easy and does not represent the modern computer vision tasks. This dataset aims to provide the accessibility of the MNIST dataset while posing a more challenging classification task.

For the experiment, we have converted the images to a column vector similar to what we have done for the MNIST dataset. We have taken 400 dilation and 400 erosion nodes in the dilation-erosion layer for this experiment. We have trained the network separately 3 times up to 300 epochs. The reported test accuracy (Table 3) is the average of the 3 runs. We see that our method gives better results.

### 4.4 CIFAR-10 DATASET

CIFAR-10 (Krizhevsky & Hinton, 2009) is natural image dataset with 10 classes. It has 50,000 training and 10,000 test images. Each of them is a color image of size $32 \times 32$. The images are converted to column vector before they are fed to the DenMo-Net. For all the networks we compare with, the experiments have been conducted with keeping the number of neurons same in the hidden layer. For maxout network each hidden neuron have two extra nodes over which the maximum is computed. In table 4 we have reported the average test accuracy obtained over three run of 150 epochs. The change of accuracy over epochs is also shown in figure 5a when number of hidden neurons is 600. As it can be seen from both the table and the figure that DenMo-Net achieves the best accuracy in all the cases. Maxout network lags behind even with more number of parameters. This

Table 3: Achieved accuracy in the test set

| Dataset | Test accuracy | |
|---|---|---|
| | DenMo-Net | State of the art |
| MNIST | 98.43 (l = 400) | **99.79** (Wan et al., 2013) |
| Fashion-MNIST | **89.87** (l = 800) | 89.70 (Xiao et al., 2017) |

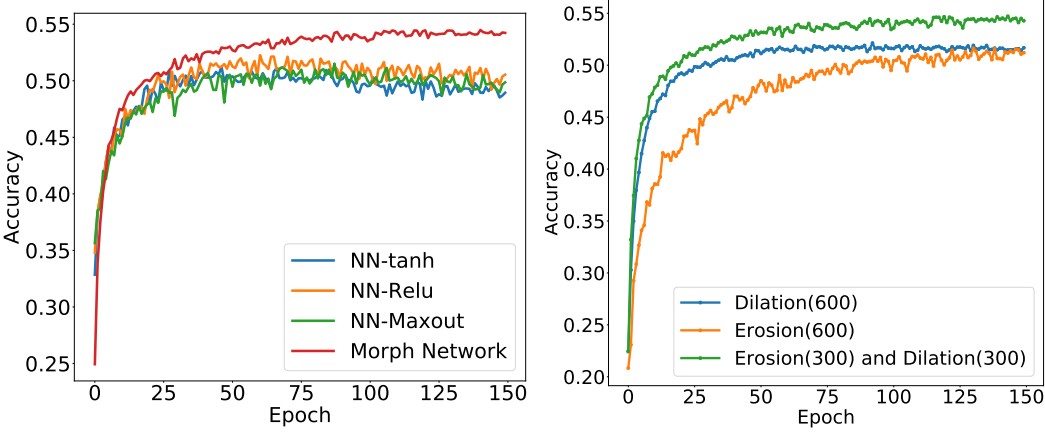

(a) On different networks (150 epochs, 600 hidden nodes)

(b) Using only dilation, only erosion and both

Figure 5: Test accuracy over epochs on CIFAR-10 dataset

Table 4: Test accuracy achieved on CIFAR-10 dataset by different networks

| Architecture | l=200 | | l=400 | | l=600 | |
|---|---|---|---|---|---|---|
| | parameters | accuracy | parameters | accuracy | parameters | accuracy |
| NN-tanh | 616,610 | 48.88 | 1,233,210 | 49.39 | 1,849,810 | 51.24 |
| NN-ReLU | 616,610 | 49.28 | 1,233,210 | 50.43 | 1,849,810 | 52.25 |
| Maxout-Network | 1,231,210 | 49.51 | 2,462,410 | 50.10 | 3,693,610 | 51.51 |
| DenMo-Net | 616,610 | **51.84** | 1,233,210 | **53.41** | 1,849,810 | **54.49** |

happens because our network is able to learn more hyperplanes with number of parameters similar to normal artificial neural networks. When using only a single type of neurons in our network, we see a different result for this dataset (Figure 5b). The network takes time to learn with only erosion neurons. The situation improves a little when using only dilation neurons. When using both type of morphological neurons, the network is able to perform better by leveraging the power of both the operations.

## 4.5 CIFAR-100 DATASET

CIFAR-100 (Krizhevsky & Hinton, 2009) is a image dataset similar to CIFAR-10 but with 100 classes with 600 images in each. There are 500 training and 100 testing images for each class. The training has been done similar to what is done for CIFAR-10. Network has been trained with batch size 100. We have reported the average test accuracy of 3 run with 100 epochs each in table 5. The change of test accuracy over the epochs is plotted in figure 6a. The results show trend similar to what is observed in other dataset. DenMo-Net is giving better result with comparable number of trainable parameters and trains much faster. When the type of neurons is restricted in our network, the results are similar to what we have seen for CIFAR-10 dataset (Figure 6b). Using only erosion neurons, the networks lags behind in terms of accuracy. But interestingly, using erosion only the network is able to perform better than using both type of morphological neurons.

## 5 DISCUSSIONS

### 5.1 GRADIENT PROPAGATION

In our network we are learning the structuring element and the weights of the linear combination layer using gradient descent method. While learning the structuring elements we may encounter two kinds of problem. Dilation/erosion operation involves max/min operation. As shown in Section 3.2, the gradient of the loss with respect to the structuring element contains a single non-zero element.

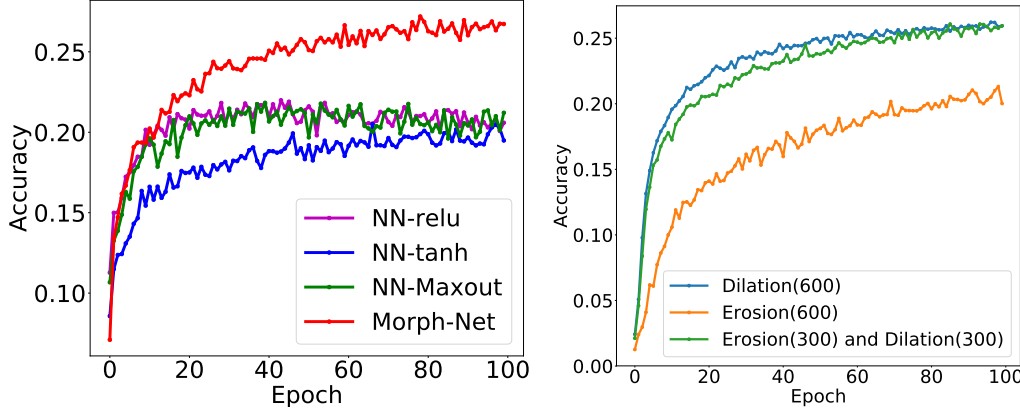

(a) On different networks (100 epochs, 500 hidden nodes)

(b) Using only dilation, only erosion and both

Figure 6: Test accuracy over epochs on CIFAR-100 dataset

Table 5: Comparison with Baseline CIFAR100

| Architecture | l=200 | | l=400 | | l=600 | |
|---|---|---|---|---|---|---|
| | parameters | accuracy | parameters | accuracy | parameters | accuracy |
| NN-tanh | 634,700 | 19.50 | 1,269,300 | 19.62 | 1,903,900 | 20.46 |
| NN-ReLU | 634,700 | 17.83 | 1,269,300 | 19.63 | 1,903,900 | 20.77 |
| Maxout-Network | 1,249,300 | 21.58 | 2,498,500 | 21.49 | 3,747,700 | 21.69 |
| DenMo-Net | 634,700 | **23.65** | 1,269,300 | **25.89** | 1,903,900 | **26.93** |

Therefore, only a single element of the structuring element is updated. So, the learning can be slow. On the other hand, some values of the structuring element may not get updated at all.

## 5.2 STACKING MULTIPLE LAYERS

We have defined the network and have shown its properties when only three layers are employed in the network. Although it may seem stacking multiple of these layers would translate to better results, the results are showing the opposite. Straight-forward stacking of the layers we have used in our network can give rise to two kinds of network.

**Type-I** Multiple dilation-erosion layer, followed by a single linear combination layer at the end.

**Type-II** Dilation-Erosion layer followed by a linear combination layer. Then another dilation-erosion layer followed by one more linear combination layer and so on.

For the network of Type-I, it can be argued that the network is performing some combination of opening and closing operation, and their linear combination. As there are dilation-erosion (DE) layers back to back, the problem of gradient propagation is amplified. As a result it takes much more time to train than single layer architecture (Table 6).

Similar explanation doesn't work for Type-II networks. From Figure 7 we see that the network has tendency to overfit. We believe its understanding requires further exploration.

## 6 CONCLUSION

In this paper we have proposed a new class of networks that uses both normal and morphological neurons. These network consists of three layers only: input layer, dilation-erosion layer with dilation and erosion neurons followed by linear combination layer giving the output of the network with normal artificial neurons. We have done our analysis using this three layer network only, but its deeper version can also be explored. We have shown that this three layer architecture can approximate any

Table 6: Test accuracy achieved on CIFAR-10 and CIFAR-100 dataset by different networks

| Network | Accuracy | |
|---|---|---|
| | CIFAR-10 | CIFAR-100 |
| NN-tanh (400D-200D-100D) | 51.6 | 24.37 |
| NN-ReLU (400D-200D-100D ) | 52.79 | 24.37 |
| DenMo-Net (400DE-100FC) | 53.41 | 25.89 |
| DenMo-Net (400DE-200DE-100FC) | 51.2 | 23.31 |
| DenMo-Net (400DE-200FC-100DE-100FC) | 39.2 | 16.32 |

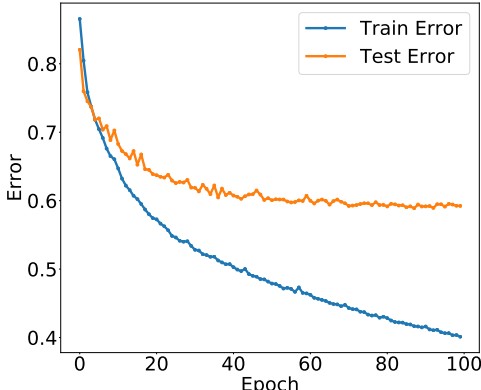

Figure 7: Train and Test accuracy achieved over epochs in the CIFAR-10 dataset

sufficiently smooth function without requiring any non-linear activation function. These networks are able to learn a large number of hyperplanes with very few neurons in the dilation-erosion layer thereby providing superior results compared to other networks with three layer architecture. The improved results could also be the result of 'feature selection' by the *max/min* operator in the dilation erosion layer. In this work we have only worked with fully connected layers, i.e. a node in a layer is connected to all the nodes in the previous layer. This type of connectivity is not very efficient for image data where architectures with convolution layers perform better. So, extending this work to the case where a structuring element operates by sliding over the whole image, should be the next logical step.

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

## APPENDIX A    PROOF OF LEMMA 1

From equation 9 we have

$$g(\boldsymbol{x}) = \sum_{i=1}^{n} \omega_i^+ z_i^+ + \sum_{j=1}^{m} \omega_j^- z_j^-. \tag{15}$$

Now this equation can be rewritten as follows

$$g(\boldsymbol{x}) = \sum_{i=1}^{n} \omega_i^+ \max_k(x_k + s_{ik}^+) + \sum_{i=1}^{m} -\omega_i^- \max_k(s_{ik}^- - x_k), \tag{16}$$

where $s_{ik}^+$ and $s_{ik}^-$ denote the $k^{th}$ component of the $i^{th}$ structuring element of dilation and erosion neurons, respectively. The above equation can be further expressed in the following form,

$$g(\boldsymbol{x}) = \sum_{i=1}^{n} \alpha_i^+ \max_k(\beta_i^+ x_k + d_{ik}^+) + \sum_{i=1}^{m} \alpha_i^- \max_k(\beta_i^- x_k + d_{ik}^-). \tag{17}$$

Where $\beta_i^+$, $\beta_i^-$, $d_{ik}^+$ and $d_{ik}^-$ are define in the following way

$$\beta_i^+ = \begin{cases} \omega_i^+ & \text{if } \omega_i^+ \geq 0 \\ -\omega_i^+ & \text{if } \omega_i^+ < 0 \end{cases} \qquad\qquad \beta_i^- = \begin{cases} -\omega_i^- & \text{if } \omega_i^- \geq 0 \\ \omega_i^- & \text{if } \omega_i^- < 0 \end{cases}$$

$$d_{ik}^+ = \begin{cases} s_{ik}^+ \omega_i^+ & \text{if } \omega_i^+ \geq 0 \\ -s_{ik}^+ \omega_i^+ & \text{if } \omega_i^+ < 0 \end{cases} \qquad\qquad d_{ik}^- = \begin{cases} s_{ik}^- \omega_i^- & \text{if } \omega_i^- \geq 0 \\ -s_{ik}^- \omega_i^- & \text{if } \omega_i^- < 0 \end{cases}$$

$$\alpha_i^+ = \begin{cases} 1 & \text{if } \omega_i^+ \geq 0 \\ -1 & \text{if } \omega_i^+ < 0 \end{cases} \qquad\qquad \alpha_i^- = \begin{cases} -1 & \text{if } \omega_i^- \geq 0 \\ 1 & \text{if } \omega_i^- < 0 \end{cases}$$

Now, without any loss of generality we can write equation 17 as follows

$$g(\boldsymbol{x}) = \sum_{i=1}^{m+n} \alpha_i \max_k(\beta_i x_k + d_{ik}) \tag{18}$$

where

$$\beta_i = \begin{cases} \beta_i^+ & \text{if } i \leq n \\ \beta_{i-n}^- & \text{if } n < i \leq m+n \end{cases} \qquad\qquad d_{ik} = \begin{cases} d_{ik}^+ & \text{if } i \leq n \\ d_{(i-n)k}^- & \text{if } n < i \leq m+n \end{cases}$$

$$\alpha_i = \begin{cases} \alpha_i^+ & \text{if } i \leq n \\ \alpha_{(i-n)}^- & \text{if } n < i \leq m+n \end{cases}$$

Finally, we can rewrite equation 18 as

$$g(\boldsymbol{x}) = \sum_{i=1}^{l} \alpha_i \phi_i(\boldsymbol{x}), \tag{19}$$

where $l = m + n$, $\alpha_i \in \{1, -1\}$ and $\phi_i(\boldsymbol{x})$'s are of the following form

$$\phi_i(\boldsymbol{x}) = \max_k(\boldsymbol{v}_{ik}^T \boldsymbol{x} + d_{ik}), \tag{20}$$

with

$$\boldsymbol{v}_{ikt} = \begin{cases} \beta_i & \text{if } t = k \\ 0 & \text{if } t \neq k \end{cases} \tag{21}$$

In equation 20, $\boldsymbol{v}_{ik}^T \boldsymbol{x} + d_{ik}$ is affine and $\alpha_i \phi_i(\boldsymbol{x})$ is a $d$-order hinge function. Hence $\sum_{i=1}^{l} \alpha_i \phi_i(\boldsymbol{x})$ i.e., $g(\boldsymbol{x})$ represents sum of multi-oder hinge function.

However, it may be noted that taking $l \geq d$ results hinge hyper planes which can span any where in $d$ dimensional input space. We can assume there are $l_1$ and $l_2$ number of terms where $\alpha = 1$ and $\alpha = -1$ respectively, then

$$g(\boldsymbol{x}) = \sum_{i=1}^{l_1} \phi_i'(\boldsymbol{x}) - \sum_{i=1}^{l_2} \phi_i''(\boldsymbol{x}), \tag{22}$$

where $l_1 + l_2 = l$ and $\phi_i'(\boldsymbol{x}), \phi_i''(\boldsymbol{x})$ is of same form as equation 20. Threfore can write,

$$\sum_{i=1}^{l_1} \phi_i'(\boldsymbol{x}) = \sum_{i=1}^{l_1} \max_k (\boldsymbol{v}_{ik}^T \boldsymbol{x} + d_{ik}), \tag{23}$$

$$\sum_{i=1}^{l_1} \phi_i'(\boldsymbol{x}) = \max_{k_1, k_2, k_3, .., k_{l_1}} ((\sum_{i=1}^{l_1} \boldsymbol{v}_{ik_1})^T \boldsymbol{x} + \sum_{i=1}^{l_1} d_{ik_i}) \tag{24}$$

where $k_i \in \{1, 2, .., d+1\} \forall i$. In equation 24 we are taking maximum of $(d+1)^{l_1}$ terms. Similarly we can derive same expression for $\sum_{i=1}^{l_2} \phi_i''(\boldsymbol{x})$ . Hence out of all $l$ number of sum we can select $d$ number of coefficient of $\boldsymbol{x}$. So, $l \geq d$ results in hinge hyperplanes which can span any where in $d$-dimensional space.

## APPENDIX B    ALTERNATIVE THEORETICAL JUSTIFICATION

Here give an alternate proof of the claim that $g(\boldsymbol{x})$ can approximate any continuous function. For that take the help of following two proposition.

**Proposition 3 (From Wang (2004))** *Any continuous piece-wise linear function (PWL) can be expressed by as difference of two convex PWL function.*

Then we show the following,

**Lemma 2** *$g(\boldsymbol{x})$ is a continuous piece-wise linear function.*

**Proof** From equation 19, without any loss of generality we can assume there are $t_1$ and $t_2$ number of terms where $\alpha = 1$ and $\alpha = -1$ respectively, then

$$g(\boldsymbol{x}) = \sum_{i=1}^{t_1} \phi_i'(\boldsymbol{x}) - \sum_{i=1}^{t_2} \phi_i''(\boldsymbol{x}), \tag{25}$$

where $t_1 + t_2 = l$ and $\phi_i'(\boldsymbol{x}), \phi_i''(\boldsymbol{x})$ are of same form as equation 20.

As sum of PWL functions is also a PWL function, hence each $\sum_{i=1}^{t_1} \phi_i'(\boldsymbol{x})$ and $\sum_{i=1}^{t_2} \phi_i''(\boldsymbol{x})$ and PWL. Now, if $t_1 > 0$, from Proposition 3 we can conclude that $g(x)$ is PWL linear function since difference of two continuous PWL function is PWL function . If $t_1 = 0$ then $g(x)$ becomes PWL concave function.Hence, can say $g(\mathbf{x})$ is PWL function.

It may be noted that if $l < d$ then PWL hyperplane will be in parallel to at least one of the axis. Taking $l \geq d$ results PWL hyperplane which may span anywhere in d dimensional space.

**Theorem 2 (Universal approximation)** *Using only a single dilation-erosion layer followed by a linear combination layer any continuous function can be approximated.*

**Sketch of Proof** From proposition 2 we get that any continuous function can be well approximated by a PWL function with an error bound of $\epsilon$. Now from lemma 2 we know that our DenMo-Net with of $n$ dilation and $m$ erosion neurons followed by a linear combination layer computes a PWL function. Hence we can say our network can approximate any continuous function. In general if we increase the neurons in the dilation-erosion layer, number of affine function in $g(\mathbf{x})$ (equation 18) increases and the error bound $\epsilon \to 0$ as the number of nodes in dilation-erosion layer increases.

