# OpenReview forum: "Dense Morphological Network: An Universal Function Approximator"
_ICLR.cc/2019/Conference_

### Official Review · AnonReviewer2 · 2018-11-02
**Interesting idea for using morphological operators but too preliminary**

**Rating:** 5
**Confidence:** 5

**Review:**

The authors introduce Morph-Net, a single layer neural network where
the mapping is performed using morphological dilation and erosion.
I was expecting something applied to convolutional networks as such operators
are very popular in image processing, so the naming is a bit misleading.

It is shown that the proposed network can approximate any smooth function,
assuming a sufficiently large number of hidden neurons, that is a nice result.

Clarity should be improved, for example it is mentioned that the structuring
element is learned but never clearly explained how and what difficulties it poses.
In the main text it is written that alpha is {-1, 1}, which would result in a
combinatorial search, but never explained how it is learned in practice.
This is shown only in the appendix but it is not clear to me that using a binarization
with the weights is not prone to degenerate solutions and/or to learn at all
if proper initialization is not used.
Did the authors experiment with smooth versions or other form of binarization with
straight-through estimator or sampling?

In the proof for theorem 1 it is not clear if the convergence of the proposed
network is faster or slower than that of a classic single layer network.

The main result of the paper is that the structuring element can be learned,
but there is no discussion on what it is learned. Also, there is no comparison
on related approaches that try to learn the structuring element in an end-to-end
fashion such as [1].

Experiments lack a more thorough comparison with state-of-the-art and at least
an ablation study to show that the proposed approach is effective and has merit.
For example, what is the relative contribution of using dilation and erosion
jointly versus either one of them.
What is the comparison with a winner-take-all unit over groups of neurons
such as max-pooling?

It seems that extending the work to multiple layers should be trivial but it is
not reported and is left to future investigations. This hints at issues with
the optimization and should be discussed, is it related to the binarization
mentioned above?

Overall the idea is interesting but the way the structuring element is learned
should be discussed in more details and exemplified visually. Experiments need
to be improved and overall applicability is uncertain at this stage.

=======
[1] Masci et al., A Learning Framework for Morphological Operators Using Counter--Harmonic Mean.

---

> ### Author Response · Authors · 2018-11-17
> **Our work is one of the forerunner work and it opens a many directions of future research[Part 1/2]**
>
> Thank you  for reviewing our paper with valuable and detailed comments.
>
> Q1: "The authors introduce Morph-Net, a single layer neural network where the mapping is performed using morphological dilation and erosion.I was expecting something applied to convolutional networks as such operators are very popular in image processing, so the naming is a bit misleading."
> >> We will update the name of the paper to "Dense Morphological Network: A Universal Function Approximator" to reduce confusion.
>
> Q2: "It is shown that the proposed network can approximate any smooth function, assuming a sufficiently large number of hidden neurons, that is a nice result.Clarity should be improved, for example it is mentioned that the structuring element is learned but never clearly explained how and what difficulties it poses."
> >> In our network we learn the structuring element and the weights of the linear combination layer using gradient descent method for minimizing loss. While learning structuring elements there maybe following problem.
> Dilation (Erosion) operation involves max (min) operation. So it implies that, during back propagation, gradient of loss function with respect to components of structuring element is zero except the component corresponding to max (min). That means for each data only one component of structuring element may be updated. As a result, the learning  process may be slow.
> We did not notice any other difficulties these dilation-erosion operation may give rise to. Note that, in this work, We have focused only on how the network works in general settings.
>
> Q3: "In the main text it is written that alpha is {-1, 1}, which would result in a combinatorial search, but never explained how it is learned in practice. This is shown only in the appendix but it is not clear to me that using a binarization with the weights is not prone to degenerate solutions and/or to learn at all if proper initialization is not used. Did the authors experiment with smooth versions or other form of binarization with straight-through estimator or sampling?"
> >> The parameter alpha is not learned in our method, not even used. It is introduced only for proving the theorems. We only learn the structuring element and the weights of the linear combination layer during the training.
>
>
> Q4: "Review: In the proof for theorem 1 it is not clear if the convergence of the proposed
> network is faster or slower than that of a classic single layer network."
> >> Theorem 1 only shows that our network can approximate any continuous function provided there are enough nodes in dilation-erosion layer. We are not claiming anything regarding the convergence rate. However, it is already mentioned that training our network may be slower at times depending on the dimension of data.

---

> > ### Author Response · Authors · 2018-11-17
> > **Our work is one of the forerunner work and it opens a many directions of future research[Part 2/2]**
> >
> > Q5: " The main result of the paper is that the structuring element can be learned, but there is no discussion on what it is learned. Also, there is no comparison on related approaches that try to learn the structuring element in an end-to-end fashion such as [1].
> > >> Thank you for the reference. we are learning the parameters of the structuring elements  in dilation-erosion layer  and the  parameters of the weighted combination layer. However, our main contribution is not the learning of structuring element. The main contribution is to design a dense morphological networks with dilation-erosion and their linear combination having similar expressive power as the artificial neural networks. As we are using morphological operations, learning of structuring elements comes into picture and for our case the size of the structuring element is same as that of the input. This is not the case for [1]. In our paper we are  flattening the image (cifar-10, cifar-100)  and producing the  class label.
> >
> > Q6: "Experiments lack a more thorough comparison with state-of-the-art and at least an ablation study to show that the proposed approach is effective and has merit. For example, what is the relative contribution of using dilation and erosion jointly versus either one of them. What is the comparison with a winner-take-all unit over groups of neurons such as max-pooling?"
> > >> Thank you for the suggestion. We have shown the contribution of dilation and erosion neurons for the toy data only. We have updated the manuscript to show this relative comparison in other data sets.
> >
> > Q7: "It seems that extending the work to multiple layers should be trivial but it is not reported and is left to future investigations. This hints at issues with the optimization and should be discussed, is it related to the binarization mentioned above? "
> > >> You are right,  it is  not that trivial to extend the work for multiple layers. Since our theoretical justification is on single layer, we have not shown the results with multiple layers.
> > However, based on the reviewer's suggestion, we have added some results with multiple layers in Section 5 and
> > Table 6.
> > Extension of this network to multiple layers can be done in two ways.
> > [Type-I] Multiple dilation-erosion layer, followed by a single linear combination layer at the end.
> > [Type-II] A layer-unit may be defined as Dilation-Erosion layer followed by a linear combination layer. Then this layer-unit may be repeated desired number of times to realize the multi-layer dense morphological network.
> > For the network of Type-I, it can be argued that the network is performing some combination of opening and closing operation, and their linear combination. As there are dilation-erosion (DE) layers back to back, the problem of gradient propagation is amplified. As a result it takes much more time to train than single layer architecture (Table 6).
> > Similar explanation doesn't work for Type-II networks. From Figure 7 we see that the network has tendency to over-fit.
> >
> > Q8: Overall the idea is interesting but the way the structuring element is learned should be discussed in more details and exemplified visually. Experiments need to be improved and overall applicability is uncertain at this stage.
> > >> Some more experimental results and explanation are incorporated in the revised version.
> >
> > Thank you again, please let us know if there are any queries or confusion.

---

### Official Review · AnonReviewer3 · 2018-11-04
**The proposed idea is to replace standard nonlinear activation function with an erosion/dilation operation. The authors report encouraging results but the baseline networks are not state-of-the-art.**

**Rating:** 5
**Confidence:** 4

**Review:**

This paper proposes to replace the standard RELU/tanh units with a combination of dilation and erosion operations, arguing for the observation that the new operator creates more hyper-planes and therefore have more expressive power.

The paper is interesting and there are encouraging results which show a couple of percentage improvements over relu/tanh units.  This paper is also clearly written and easy to understand. However there are two issues:
1. It is somewhat unclear from the paper what  is the main novelty here (compared to existing morpho neurons), is it the learning of the structuring element s? is it the combination of the dilation+erosion operations?
2. The second issue is that presumably due to the fact that Conv layers are not used, the accuracy on cifar-10 and cifar-100 are significantly lower than state-of-the-art. It would make the paper extremely strong if the improvement translated to CNNs which are performing near the state-of-the-art. What happens if relu units in CNNs were swapped out for the proposed dilation/erosion operators?

---

> ### Author Response · Authors · 2018-11-12
> **The main idea is to replace the normal artificial neural networks by using basic morphological operations. Non-requrement of activation functions is a by-product.**
>
> Thank you  for reviewing our paper with valuable and detailed comments.
>
> Q1: "This paper proposes to replace the standard RELU/tanh units with a combination of dilation and erosion operations, arguing for the observation that the new operator creates more hyper-planes and therefore have more expressive power"
> >> In this paper we propose to build networks with basic morphological operations. This gives us the power to build networks with similar expressive power of the normal artificial neural networks without the need of activation functions, while {\em requiring less number of parameters}. Replacing the standard nonlinear activation function is not the main goal, it may be a by-product of the dilation-erosion operation. However, thanks for pointing this out.
>
>
> Q2: "The paper is interesting and there are encouraging results which show a couple of percentage improvements over relu/tanh units.  This paper is also clearly written and easy to understand. However there are two issues:
> 1. It is some what unclear from the paper what is the main novelty here (compared to existing morpho neurons), is it the learning of the structuring elements? is it the combination of the dilation+erosion operations?"
> >>The main contribution are as follows.
> 1. The use of linear combination operation after dilation-erosion operation. This structure, as shown in Section 3.3, can approximate any continuous function given enough dilation/erosion neurons.
> 2. We have shown that the networks build with such layers do not need activation functions.
> 3. The use of dilation-erosion layer followed by linear combination layer greatly increases number of possible decision boundaries. As a result, complex decision boundaries can be learned using small number of parameters. This is visually shown using a toy dataset in Section 4.1.
>
> Note that, in the dilation and erosion layers, we have considered structuring elements only of same size. However, in the training process we learn the values of the structuring element pixels as well as the weights of the linear combination layer.
> However, we will add a paragraph highlighting our contribution in the revised version.
>
> Q3: "The second issue is that presumably due to the fact that Conv layers are not used, the accuracy on cifar-10 and cifar-100 are significantly lower than state-of-the-art. It would make the paper extremely strong if the improvement translated to CNNs which are performing near the state-of-the-art. What happens if relu units in CNNs were swapped out for the proposed dilation/erosion operators?"
> >> It is true that the convolution layers perform well for images as they are able to extract features based on spatial information.
> However, in this work we have defined our network for flattened input data. Our network structure and the operations is totally different than classical neural network. For instance in the first layer we take addition (subtraction) with weights (i.e., values of structuring element) instead of multiplication and then take max (min) instead of addition to implement 1-D dilation (erosion) operation. In the next layer we are taking weighted combination of the output from this layer.
> So, we do not and cannot directly use convolution layer in our network, and just swapping the activation function with dilation/erosion layers will not work. For this reason, we have compared our work with neural networks containing dense layers. For harnessing the spatial information, 2D dilation-erosion layer may be defined where the structuring element is much smaller than the input (image).
>
> Thank you again, please let us know if there are any queries or confusion.

---

> > ### Comment · AnonReviewer3 · 2018-11-29
> > **response to answers from authors**
> >
> > Thanks for clarifying this reviewer's questions.
> > I think the novelty of linear combination of erosion/dilation units' outputs, which can approximate any continuous distribution, is novel but not substantially novel enough by its self. Ideally, it would be interesting to see empirical support from strong empirical results.
> >
> > While it is completely valid that the proposed method is not necessarily better than CNNs designed for image recognition tasks, would it be possible to compare to tasks where dense fully connected nets achieves state-of-the-art and show the effects (performance) of swapping in the proposed dilation/erosion network?

---

> > > ### Author Response · Authors · 2018-12-01
> > > **Thank you for the update**
> > >
> > > Thank you for the update.
> > > It would be nice if we get some dataset names, where dense networks achieve state-of-art performances.

---

### Official Review · AnonReviewer4 · 2018-11-12
**A nice idea but weak empirical results**

**Rating:** 5
**Confidence:** 3

**Review:**

* Update:
Thanks for you answer and clarification. While the Morph-net appears novel, the authors only report result for image classification task and don't achieve as good performance as standard convolutional baselines. Given the current empirical evaluation, I find hard to assess how significant is the contribution. I would encourage the authors to either compare on a task where dense networks achieve state-of-art performances or extend their approach to 2D inputs.


* Review

This paper introduces Morph-Net, a new architecture that intertwine morphological operator such as dilation/erosion with linear layer. Authors first show than Morph-Net are universal approximator. Morph-Net can be expressed as a sum of multi-order hinge functions which can approximate any continuous function. They then validate empirically the Morph-Net on  MNIST, FashionMNIST,  CIFAR10 and CIFAR100 datasets. In particular, authors investigate a 3 layers  fully-connected Morph-Net and shows that it can outperform its Tanh/Relu/Maxout counterparts.

The paper is a nice read also some specific point could be clarify. For instance it is not clear how the structuring elements of the dilation/erosion are learned? Are the learned simply through backpropagation? Also, it is not clear to me how Morph-Net differs from the previously proposed morphological neurons?

Empirical evaluation of Morph-Net could be improved as well. In particular, authors focus on image classification task. While they show that Morph-Net can outperform other fully connected architecture, the results on CIFAR10/100 seems low compared to convolutional network. It raises the question of the advantages of Morph-Net over convolutional neural networks ?  Authors also limit their exploration to  3-layer networks. Why don’t you explore deeper network for both baseline and Morph-Net?  Finally, if I am not mistaken, authors use the same set of hyperparameters for the baselines/Morph-Net? It is not clear to me if the hyperparameters are optimal for all the approach? They might give an unfair advantage to one of the baseline or Morph-Net?

Overall, this paper present a nice idea. Showing the Morph-Net is an universal approximator is a nice result. However, the empirical evaluation could be improved. It is not clear to me at this point if Morph-Net brings a benefit compare to convolutional net for image classification task.

---

> ### Author Response · Authors · 2018-11-17
> **Weak but with respect to deep CNNs.**
>
> Thank you  for reviewing our paper with valuable and detailed comments.
>
> Q1: "The paper is a nice read also some specific point could be clarify. For instance it is not clear how the structuring elements of the dilation/erosion are learned? Are the learned simply through backpropagation? "
> >> You are right, the structuring elements in dilation-erosion layer and the weights of linear combination layer learn through back propagation. we have added a sub section highlighting a gradient calculation and back-propagation on our network.
>
>
> Q2: "Also, it is not clear to me how Morph-Net differs from the previously proposed morphological neurons?"
> >> Morphological neurons have been defined in the literature in different ways. Although, all of them use dilation and erosion operation, this is usually followed by an additional operation (e.g. activation function [Ritter and Sussner 1996]). For our network we have defined dilation and erosion neurons that perform only dilation and erosion operation respectively. Apart from that, our network also employs an additional linear combination layer. As shown in Theorem 1, these two layers together can approximate any smooth continuous function without requiring additional activation functions. This claim cannot be made if only morphological neurons are used in the network.
>
>
> Q3: "Empirical evaluation of Morph-Net could be improved as well. In particular, authors focus on image classification task. While they show that Morph-Net can outperform other fully connected architecture, the results on CIFAR10/100 seems low compared to convolutional network.It raises the question of the advantages of Morph-Net over convolutional neural networks ?"
> >> It is true that the convolution networks perform well for images as they are able to extract features based on spatial information. However, in this work we have defined our network for flattened input data and densely connected layers. For this reason our network does not have the advantage of conv type of operation. The main aim is to show that this type of network have capabilities similar to artificial neural networks while using less number of parameters. The advantage of this network over CNNs can possibly be shown after defining the dilation and erosion as 2D operations.
>
> Q4: "Authors also limit their exploration to  3-layer networks. Why don’t you explore deeper network for both baseline and Morph-Net?"
> >> We have proved that using only 3-layer (considering input, dilation-erosion and linear combination layers) network any continuous function can be approximated. That is why we have shown the results using 3-layer networks only. As for going to the multi-layer case, the layers can be stacked in two ways.
> [Type-I] Multiple dilation-erosion layer, followed by a single linear combination layer at the end.
> [Type-II] A layer-unit may be defined as Dilation-Erosion layer followed by a linear combination layer. Then this layer-unit may be repeated desired number of times to realize the multi-layer dense morphological network.
> For the network of Type-I, it can be argued that the network is performing some combination of opening and closing operation, and their linear combination. As there are dilation-erosion (DE) layers back to back, the problem of gradient propagation is amplified. As a result it takes much more time to train than single layer architecture (Table 6).
> Similar explanation doesn't work for Type-II networks. From Figure 7 we see that the network has tendency to overfit.
>
>
> Q5: "Finally, if I am not mistaken, authors use the same set of hyperparameters for the baselines/Morph-Net? It is not clear to me if the hyperparameters are optimal for all the approach? They might give an unfair advantage to one of the baseline or Morph-Net?"
> >> Yes, we have used same hyperparamenters for the baseline and Morph-Net, because we want to show that our network is more expressive when using similar hyper-parameters. The hyperparameters may not be optimal for any of the network. This is done for comparison purpose only.
>
> Q6: "Overall, this paper present a nice idea. Showing the Morph-Net is an universal approximator is a nice result. However, the empirical evaluation could be improved. It is not clear to me at this point if Morph-Net brings a benefit compare to convolutional net for image classification task"
> >> Since we have not defined 2D Dilation/Erosion in this paper so we refrain ourselves from commenting on this issue, i.e., whether Morph-Net brings a benefit compare to Convolutional or not. However we believe, this is one of the forerunner work and it opens a many directions of future research.
>
> Thank you again, please let us know if there are any queries or confusion.

---

> > ### Author Response · Authors · 2018-12-01
> > **Thank you for the update**
> >
> > Thank you for the update.
> >
> > Modifying the network to accept 2D input is straightforward. But the theorem we have proved for the dense single layer case will not hold there. On the other hand, if we use 2D morphological operations in the network a single hidden layer will not be sufficient. So, we have to extend the network to the multi layer case. But then the same problems(case 1. over fitting;  case2: slow learning) of the dense multi layer case arises in this situation also.
> >
> > However, as I asked reviewer3, It would be nice If we get some dataset names, where dense networks achieve state-of-art performances.

---

### Public Comment · ~Elad_Eban1 · 2018-10-05
**Name conflict**

Hi,

It would be nice and very useful if you consider renaming your paper, as a paper named "MorphNet: Fast & Simple Resource-Constrained Structure Learning of Deep Networks", was published in CVPR 2018. I believe this is  bad name conflict as the papers topic are related enough to cause confusion.

Respectfully,

Elad Eban

see:
https://arxiv.org/abs/1711.06798

---

> ### Author Response · Authors · 2018-10-07
> **Thank you for  pointing out**
>
> Thank you for pointing out. We will change the title to something else.
>
> Thank you,

---

### Author Response · Authors · 2018-11-17
**Revision**

- Ablation study on other data set
- Discussion on Multi-layer and gradient propagation is added.
- [1] Ref added
- Title has been changed from "Morph-Net" to "DENSE MORPHOLOGICAL NETWORK"
- Our contribution is highlighted in introduction

---

### Meta-Review · Area_Chair1 · 2018-12-17
**Interesting ideas that requires to more adequate baselines**

**Confidence:** 5
**Recommendation:** Reject

**Metareview:**

This work presents an interesting take on how to combine basic functions to lead to better activation functions. While the experiments in the paper show that the approach works well compared to the baselines that are used as reference, reviewers note that a more adequate assessment of the contribution would require comparing to stronger baselines or switching to tasks where the chosen baselines are indeed performing well. Authors are encouraged to follow the many suggestions of reviewers to strengthen their work.